# Trends in Frailty and Use of Evidence-Based Pharmacotherapy for Heart Failure in Australian Hospitalised Patients: An Observational Study

**DOI:** 10.3390/jcm10245780

**Published:** 2021-12-10

**Authors:** Yogesh Sharma, Chris Horwood, Paul Hakendorf, Campbell Thompson

**Affiliations:** 1Department of General Medicine, Division of Medicine, Cardiac & Critical Care, Flinders Medical Centre, Adelaide 5042, Australia; 2College of Medicine and Public Health, Flinders University, Adelaide 5042, Australia; 3Department of Clinical Epidemiology, Flinders Medical Centre, Adelaide 5042, Australia; Chris.Horwood@sa.gov.au (C.H.); Paul.Hakendorf@sa.gov.au (P.H.); 4Discipline of Medicine, The University of Adelaide, Adelaide 5005, Australia; Campbell.Thompson@adelaide.edu.au

**Keywords:** heart failure, frailty, heart failure specific medications, hospitalised patients, mortality, readmissions

## Abstract

Frailty increases morbidity and mortality in heart failure (HF) patients. Current risk-adjustment models do not include frailty-status and the relationship between frailty and pharmacotherapy is unclear. This study explored trends in frailty over time and its relationship with prescription of heart failure specific pharmacotherapy in hospitalised HF patients. We used the Hospital Frailty Risk Score (HFRS) to determine frailty status of patients ≥18 years admitted between 2015–2019 at two tertiary hospitals in Australia. Patients with an HFRS ≥ 5 were classified as frail. In the 3706 patients with a mean (SD) age of 76.1 (14.4) years, 876 (23.6%) were classified as frail. HFRS was weakly correlated with age (r = 0.16) and Charlson-index (r = 0.35) (both *p* values < 0.001). Whilst frailty was more common in older HF patients (28.9% of patients ≥80 years), 15.1% of patients ≤65 years of age were also found to be frail. The proportion of frail patients increased from 19.4% in 2015 to 29.2% in 2019 despite no significant change in age during this period. The proportion of patients who received heart failure specific pharmacotherapy decreased from 86.7% in 2015 to 82.9% in 2019 (*p* value = 0.03) and frail patients were significantly less likely to be prescribed HF specific pharmacotherapy than non-frail patients (77.4% vs. 85.9%, *p* < 0.001).

## 1. Introduction

Frailty is an important prognostic factor among patients with heart failure because it increases both morbidity and mortality [1]. The current risk adjustment models determining prognosis in heart failure, however, does not incorporate frailty [2,3]. This may be because frailty assessment requires face-to-face interaction and some frailty assessment tools require special physical measurements, such as hand grip strength or a measure of gait speed [4,5]. The Hospital Frailty Assessment Tool (HFRS) has recently been developed in the UK, and can be generated based on routinely available hospital administrative data [6]. Although this tool was originally developed for individuals aged 75 years or older, it has been validated and is generalisable in different age groups and health care settings [7,8].

A number of studies [9,10,11,12] have demonstrated that despite benefits of targeted therapy among heart failure patients, the proportion of patients receiving such therapy has changed little in the last few years. Evidence [9,10] suggests that older patients, those with more frequent hospitalisations, renal insufficiency, lower blood pressure and a greater severity of heart failure (NYHA class 3 and 4) are less likely to receive any (let alone achieve targeted dose of) heart failure specific medications. It is often assumed that the reason for non-prescription or non-achievement of the targeted dose of heart failure specific medications in these patients is due to an increasing age or the higher risk of frailty in these patients [13]. Recent studies [3,10] investigating prescription practices among heart failure patients have emphasised the need for frailty assessment. Thus far, only one study [3] has investigated the trends in frequency of frailty among hospitalised heart failure patients. To our knowledge, there has been no study in the Australian health care settings which has investigated the trends in frailty over time or the prescription of heart failure specific pharmacotherapy with regards to each patient’s frailty status. This study therefore was designed to examine the prevalence and the influence of frailty over time in a cohort of patients admitted to hospital with a diagnosis of heart failure. In particular, this study examined the outcomes of admissions of these patients and the association between frailty and prescription of heart failure medications.

## 2. Materials and Methods

This retrospective study included data of all patients ≥18 years of age who were hospitalised with heart failure over a period of five years at two tertiary teaching hospitals, Flinders Medical Centre (FMC) and Royal Adelaide Hospital (RAH) in Adelaide, Australia. The ethical approval for this study was granted by the Southern Adelaide Human Research Ethics Committee. We identified all adult hospital admissions, between 1 January 2015 and 31 December 2019, with a primary diagnosis of heart failure by using the International Classification of Diseases Tenth Revision Australian Modification (ICD-10-AM) code 150, which has been previously used to define heart failure [14]. In cases where patients had multiple presentations for heart failure during the study period, then only the first admission was included.

The HFRS were calculated according to the criteria defined by Gilbert et al. [6]. HFRS is based upon administrative data by allocating point values for any of 109 select ICD codes as defined in the original publication. These codes include diagnoses such as falls, osteoporosis, spinal compression fractures, blindness, skin ulcers, delirium/dementia, Parkinson’s disease, urinary incontinence, urinary tract infections, disorders of electrolytes, drugs/alcohol abuse and sequelae of stroke such as hemiplegia and dysphagia. None of the ICD-10 codes used for the generation of the HFRS score is for heart failure, atrial fibrillation or coronary artery disease (CAD). Higher HFRS scores indicate a greater severity of frailty, and these scores can further be categorised into 3 groups: >15 (high risk), 5–15 (intermediate risk) and <5 (low risk). For this study, we classified patients with an HFRS score ≥5 as frail and those with HFRS scores of <5 as non-frail as has been conducted in previous studies [3,6]. We examined trends in frailty status at admission by determining the median HFRS scores and the proportion of frail patients according to their age group. Trends in frailty status, as determined by the HFRS over 5 years, were examined using linear regression on corresponding year for means of continuous variables, quantile regression on corresponding year for medians of continuous variables and the use of the Cochran–Armitage trend test [15] for proportion of frail patients. We used Pearson correlation to test association between the HFRS and age, Charlson comorbidity index (CCI) and length of hospital stay (LOS), and Spearman’s correlation was used for heart failure specific medications (beta blockers, angiotensin converting enzyme inhibitors (ACEi)/angiotensin receptor blockers (ARBs) and mineralocorticoid receptor antagonists (MRA).

The outcomes examined included: LOS, in hospital mortality, 30-day mortality (including in hospital mortality), 180-day mortality, 30-day readmissions and placement in a nursing home. We used multilevel mixed effects Poisson and logistic regression models to determine association between the outcome variables and frailty after adjustment for age, sex, CCI, creatinine, troponin, brain natriuretic peptide (BNP), albumin and c-reactive protein (CRP) levels. We also determined whether inclusion of the HFRS provides any additional information above and beyond that provided by age, sex, CCI and LACE scores when predicting clinical outcomes. The LACE score [16] gives a maximum of 19 points for hospital LOS (L), acuity of presentation (A), Charlson index (C) and emergency room visits in the previous 6 months (E) and has been validated for predicting readmissions or death within 30 days of hospital discharge. In addition, we also analysed outcomes in three predefined age categories (<65 years, 65–79 years and ≥80 years). The trends in outcomes over time were also analysed according to the three frailty categories.

## 3. Results

There were 5191 admissions with heart failure between 1 January 2015 and December 2019. After omitting multiple admissions, 3706 patients remained in the dataset (Figure 1). The mean age was 76.1 (14.4) years, range 19–105 years and 51.7% were males. The mean (SD) HFRS score was 3.2 (3.8) and 876 (23.6%) patients were found to be frail (HFRS ≥ 5). HFRS defined frailty was more likely to be associated with an older age, male sex, a higher CCI, increased creatinine and CRP levels but with lower haemoglobin and albumin levels (Table 1). Frailty was, however, not found to be associated with BNP or troponin levels. The HFRS was only weakly correlated with age (r = 0.16, *p* < 0.001) and CCI (r = 0.35, *p* < 0.001) in these patients.

The median (IQR) HFRS increased from 1.8 (0 to 4.1) in 2015 to 2.6 (0 to 5.5) in 2019 (trend *p* value < 0.001) although there was no significant change in median (IQR) age during this period 80.0 (69, 86) years in 2015 vs. 80.0 (68, 86) years in 2019 (*p* = 0.092). The proportion of patients who were classified as frail according to the HFRS increased from 19.4% in 2015 to 29.2% in 2019 if all age groups were combined and from 21.3% to 31.3% in those over the age of 65 years (Figure 2 and Figure 3). As expected, frailty was more common in older patients (28.9% of patients 80 years or older and 20.2% of patients between 65–79 years); however, even in patients who were younger than 65 years, a significant proportion (15.1%) of inpatients with heart failure met the HFRS case definition for frailty and the proportion of patients with frailty increased in all age groups between 2015 to 2019 (*p* < 0.001) (Figure 2 and Figure 3).

The LOS was significantly prolonged among patients who were identified as frail according to the HFRS when compared to the non-frail group (Table 2). LOS was moderately correlated with HFRS (r = 0.44, *p* < 0.001). The risks of 30-day and 180-day mortality were both significantly higher in frail patients and in all three age groups (Table 2). Frailty was associated with worse clinical outcomes in terms of LOS, 30-day and 180-day mortality in both younger patients (<65 years) and older patients (≥65 years) when compared to those who were classified as non-frail (*p* < 0.05). When compared to non-frail patients, only those frail patients who were over the age of 65 years were more likely to be admitted to a nursing home (*p* < 0.001). However, 30-day readmissions were not significantly different according to the categorisation of frailty by the HFRS in different age groups (Table 2). Multilevel regression analysis suggested that frail patients as defined by the HFRS had a 1.7-fold increased risk of prolonged LOS (IRR 1.73, 95% CI 1.65–1.82, *p* value < 0.001) and a 4-fold increased risk of predicting 30-day mortality (IRR 4.12, 95% 2.71–6.27, *p* < 0.001) after adjustment for age, sex, CCI, creatinine, BNP, troponin, CRP and albumin levels. However, HFRS was not found to predict 30-day readmissions after adjusted analysis (IRR 0.99, 95% CI 0.96–1.03, *p* value = 0.915).

In this study, the ability to predict 30-day mortality in all hospitalised heart failure patients using age, sex, CCI and LACE score was good (c-statistic 0.74). Adding the HFRS to the above model improved its ability to predict 30-day mortality (c-statistic 0.80). Similarly, the model containing age, sex, CCI, LACE score and HFRS was a better predictor of prolonged LOS (LOS > 5 days) (c-statistics 0.79) than the model without HFRS (c-statistic 0.74). However, the addition of the HFRS to the model containing age, sex, CCI and LACE did not improve the prediction for 30-day readmissions (c-statistic 0.57 without HFRS vs. 0.57 with HFRS).

Overall, 3110 (83.9%) patients received one or more heart failure specific medications (beta blockers, ACEi/ARB and MRA). However, heart failure specific medications were less likely to be prescribed to frail patients when compared to the non-frail group (77.4% vs. 85.9%, *p* value < 0.001). Patients who were categorised as non-frail according to the HFRS were more likely to receive beta blockers, ACE inhibitors and MRAs than those who were in the frail group (*p* value < 0.001) (Table 1). Similarly, a significantly higher proportion of patients who were non-frail received aspirin, anticoagulants and statins when compared to non-frail patients (*p* value < 0.001). However, there was no difference in the use of ARBs and digoxin between the frail and non-frail patients (Table 1). Both the HFRS and age were found to be negatively correlated with the use of heart failure targeted pharmacotherapy, although the correlation was weak (Spearman correlation coefficients −0.09 vs. −0.16, both *p* values < 0.001). Nevertheless, there was no correlation between Charlson index and use of heart failure specific pharmacotherapy (Spearman correlation coefficients (−0.02, *p* value = 0.467). Over the duration of this study, the proportion of patients who received heart failure specific therapy decreased from 86.7% in 2015 to 82.9% in 2019 (*p* value = 0.03) (Figure 4). However, although not reaching statistical significance, this trend was more apparent among frail patients (86% in 2015 vs.76% in 2019, *p* = 0.173) when compared to non-frail patients (87% in 2015 vs. 86% in 2019, *p* value = 0.247).

## 4. Discussion

This study proved that, over a wide age range, a significant proportion of patients who were admitted with heart failure were identified as frail according to the HFRS and the proportion of inpatients with heart failure who were frail has increased in the last five years despite no significant aging of this population. Frailty as identified by the HFRS was only modestly correlated with age and CCI. HFRS predicted LOS, mortality and admissions to nursing home but not 30-day readmissions and the addition of HFRS increased prediction of clinical outcomes such as LOS and death above and beyond age, sex, CCI and LACE score. Frail heart failure patients were less likely to receive heart failure specific therapy when compared to non-frail patients with a trend towards decreased rate of prescription over the five years of the study period.

Although increasing HFRS scores were positively correlated with age and CCI, these associations were relatively weak. HFRS does reflect aspects of frailty, which are not captured by age and CCI. This is not surprising because HFRS is based on primary and secondary diagnostic codes, which reflect conditions such as falls, sarcopenia, incontinence and urinary tract infections; none of which is captured by the CCI [6].

Our study also shows that the addition of the HFRS to standard risk prediction models [2] for post discharge outcomes provided modest additional information in terms of predicting LOS and mortality but not readmissions. These findings are similar to a Canadian study [3], which included 26,266 hospitalised heart failure patients with a mean age of 77.4 years and found that addition of the HFRS only poorly predicted outcomes in the first 30-days after discharge from hospital. Similarly, an English Longitudinal Study on Ageing (ELSA) [17], which included 5294 patients over the age of 60 years, also found that the addition of frailty scores provided only marginal improvements in predicting clinical outcomes above chronological age for several different long-term outcomes. This may be because the HFRS is a cumulative deficit model of frailty and does not capture functional status and does not correlate with measures such as hand grip strength and gait speed—which are typical phenotypic manifestations of frailty [18]. In addition, HFRS does not capture transient fluctuations induced by deconditioning associated with an acute hospital admission and also does not include variables, such as social support and socioeconomic status, which can influence outcomes such as unplanned hospital readmissions.

This study found that although there was a small change in age, the proportion of frail hospitalised heart failure patients increased between 2015 and 2019. This increment in frailty over time, although small, was statistically significant only in older patients (≥65 years). This could be related to an increase in the non-cardiovascular comorbidity burden of heart failure patients as has been reported by a number of recent studies [19,20]. This study also found that a significant proportion (15.1%) of patients younger than 65 years of age met the HFRS case definition for frailty and, similar to the older counterparts, frailty was associated with worse outcomes in the younger age group. These results are similar to a study by McAlister et al. [3], who also found that up to a quarter of hospitalised heart failure patients under the age of 65 years may be frail. This finding indicates that frailty should be considered in younger patients admitted with heart failure and not just in older patients.

This study also found that frail patients are less likely to receive evidence-based heart failure specific therapy than non-frail patients and, over time, this disparity worsened. This trend over time was evident even in older patients who were classified as non-frail by the HFRS. Our data indicate that that there is only a weak correlation between the HFRS defined frailty and age and both HFRS and age are negatively correlated with the prescription of heart failure specific medications. These findings echo results of a recent Canadian study [3], which also found a weak correlation between HFRS and age (r = 0.20, *p* < 0.001) and also found that both HFRS and age were negatively correlated with the prescription of heart failure specific therapies (r = –0.14 and r = –0.23, respectively, *p* < 0.05). It is possible that, in the absence of an objective assessment of frailty, clinicians rely on age as a proxy for frailty, so as to determine their potential suitability for receipt of heart failure specific pharmacotherapy, an assumption which has been proven to be wrong according to our study as well as others [3]. Since frail patients are at a high risk of adverse health outcomes, such as death or readmission [21,22], the potential absolute benefits of treatment may be greatest in this subgroup of patients [23].

Current guidelines from the European Society of Cardiology (ESC) and Heart Failure Association (HFA) [24] recommend heart failure specific pharmacotherapy irrespective of age. However, our study found that a large proportion of frail patients received sub-optimal heart failure therapy. In fact, recommended heart failure therapy was less likely to be prescribed to patients who had a greater severity of frailty. The potential reasons could be a real or perceived greater likelihood of side-effects or intolerance to medications in frail patients. The other reason for physicians’ inertia in prescribing medications in older frail patients could be due to a paucity of evidence because older patients have been systematically excluded from heart failure trials. This is evident from a recent study [22], which highlighted that after applying inclusion and exclusion criteria for frailty in the PARADIGM-HF, 83% of frail patients would have been excluded compared to 65% of non-frail patients (*p* < 0.001). Similarly, in the recent DAPA-HF trial [25], which investigated the benefits of Dapagliflozin in heart failure, only a minority of patients (24%) were ≥75 years and this trial also excluded patients with severe chronic kidney disease (CKD) (GFR < 30 mL/min/1.73 m^2^) and the presence of comorbidities, which could limit life expectancy. These findings suggest that it is difficult for clinicians to ascertain the benefits of heart failure specific therapy in older frail subjects.

### Limitations

We were unable to procure echocardiogram results of our patients and hence were unable to distinguish the impact of frailty in patients with heart failure with reduced and preserved ejection fractions. Nonetheless, the severity of heart failure in these patients was judged from the BNP levels [26]. We have no data on the clinical presentation of these patients nor on the presence of any contraindications to evidence-based heart failure medications and thus cannot comment upon the appropriateness of therapy. Due to the observational design of this study and the possibility of residual confounding, we cannot conclusively state the underlying reasons for the observed associations. Finally, we cannot rule out any change in the depth of coding over years in our hospitals because it has been found that hospitals with better depth of coding had a higher proportion of patients meeting the HFRS case definition of frailty.

## 5. Conclusions

This study has found that HFRS defined frailty has increased in the last five years in hospitalised heart failure patients in Australia and HFRS reflects aspects of frailty not determined by age or Charlson index. The addition of HFRS to standard risk prediction models comprising of age and Charlson provides only marginal improvements in the prediction of clinical outcomes. The prescription of heart failure targeted therapy remains poor, especially in frail hospitalised patients, and has deteriorated in the last five years. Future trials in heart failure patients should include older frail patients and the assessment of frailty should be routinely included in the management of heart failure patients because frail patients, irrespective of their age, have poorer outcomes and may gain greater benefits from heart failure targeted therapies than non-frail patients.

## Figures and Tables

**Figure 1 jcm-10-05780-f001:**
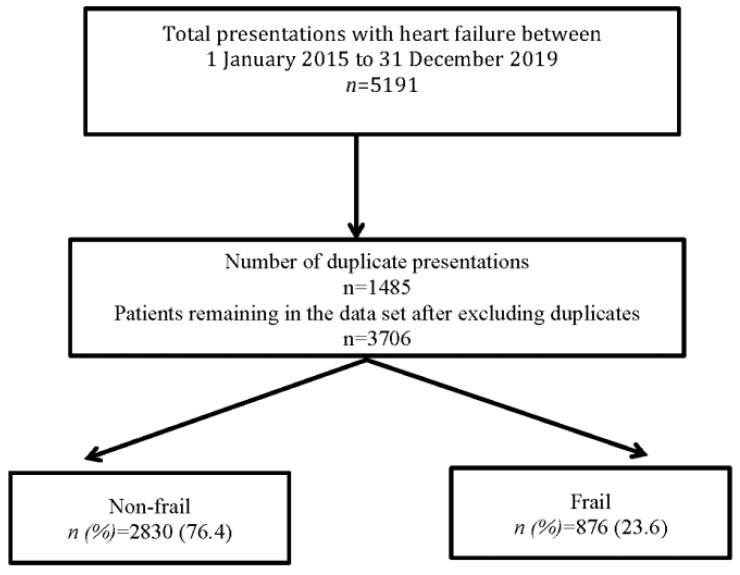
Study flow diagram.

**Figure 2 jcm-10-05780-f002:**
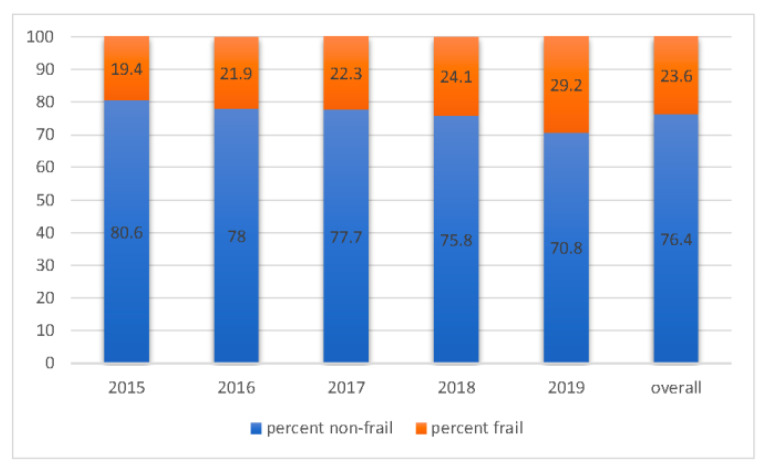
Temporal trends in frailty as defined by the Hospital Frailty Risk Score (HFRS) among heart failure patients from 2015 to 2019.

**Figure 3 jcm-10-05780-f003:**
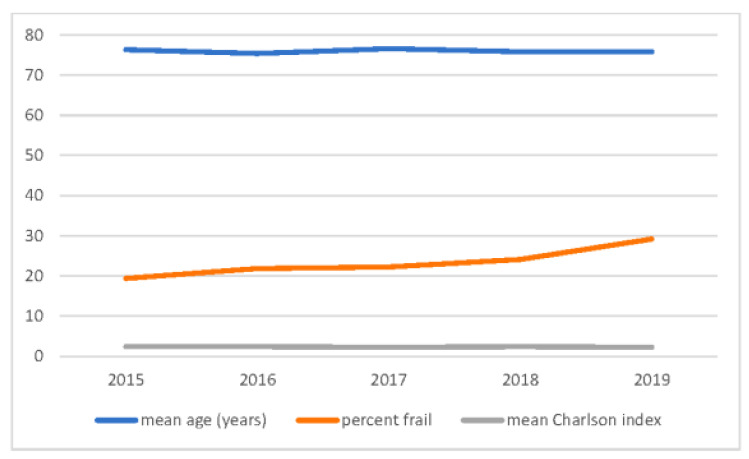
Graph showing trends in age, frailty status and Charlson comorbidity index over the duration of the study.

**Figure 4 jcm-10-05780-f004:**
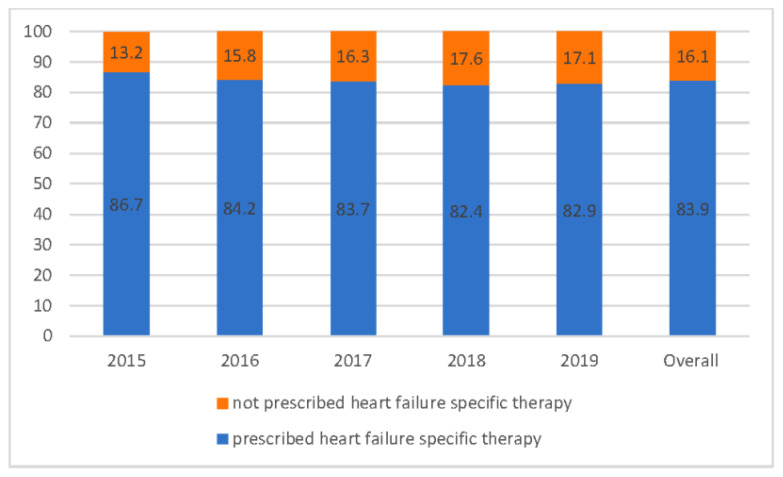
Trends in prescription of heart failure specific medications from 2015–2019.

**Table 1 jcm-10-05780-t001:** Baseline characteristics and use of heart failure specific medications according to frailty status.

Variable	Low Risk HFRS	Intermediate Risk HFRS	High Risk HFRS	*p* Value
	(HFRS < 5)	(HFRS 5–15)	(HFRS > 15)	
	*n* = 2830	*n* = 817	*n* = 59	
Age years mean (SD)	75.2 (14.4)	79.4 (12.3)	82.5 (11.1)	<0.001
Age group *n* (%)				
<65 years	598 (21.1)	101 (12.4)	5 (8.5)	<0.001
65–79 years	902 (31.9)	219 (26.8)	9 (15.2)	
≥80 years	1330 (47.0)	497 (60.8))	45 (76.3)	
Sex male *n* (%)	1463 (51.7)	414 (50.7)	40 (67.8)	0.039
HFRS mean (SD)	1.5 (1.5)	8.0 (2.5)	18.0 (2.7)	<0.001
CCI mean (SD)	2.1 (1.5)	3.3 (1.9)	3.9 (1.8)	<0.001
Creatinine mmol/L mean (SD)	114.6 (67.4)	155.7 (92.5)	172.1 (92.3)	<0.001
Haemoglobin g/L mean (SD)	124.2 (20.8)	118.1 (22.2)	118.5 (23.1)	<0.001
BNP mean ng/L (SD)	1050.9 (1514.8)	2040.0 (2076.1)	1020.5 (1030.7)	0.468
Troponins ng/L mean (SD)	1.7 (31.3)	0.6 (7.8)	0.7 (3.9)	0.567
CRP mg/L mean (SD)	21.6 (32.0)	33.1 (45.3)	35.8 (41.8)	<0.001
Albumin g/L mean (SD)	34.0 (4.9)	32.3 (5.5)	31.6 (3.9)	<0.001
MUST scores	0.5 (0.9)	0.7 (1.1)	0.8 (1.1)	0.017
Beta blockers *n* (%)	1813 (64.1)	460 (56.3)	23 (38.9)	<0.001
ACE inhibitors *n* (%)	1295 (45.8)	287 (35.1)	17 (28.8)	<0.001
ARB *n* (%)	402 (14.2)	107 (13.1)	8 (13.6)	0.720
MRA *n* (%)	1350 (40.6)	301 (36.9)	19 (32.2)	0.005
Sacubitril/Valsartan *n* (%)	53 (1.9)	11 (1.4)	0	0.352
Aspirin *n* (%)	1084 (38.3)	265 (32.4)	19 (32.2)	0.007
Warfarin *n* (%)	501 (17.7)	192 (23.3)	12 (20.3)	0.001
DOACs *n* (%)	648 (22.9)	171 (20.9)	8 (13.6)	0.131
Digoxin *n* (%)	398 (14.1)	136 (16.6)	11 (18.6)	0.128
Statins *n* (%)	1459 (51.6)	373 (45.7)	17 (28.8)	0.001
SGLT2 inhibitors *n* (%)	53 (1.9)	17 (2.1)	1 (1.7)	0.922
Ivabradine *n* (%)	68 (2.4)	17 (2.1)	1 (1.7)	0.821

HFRS, hospital frailty risk score; SD, standard deviation; CCI, Charlson comorbidity index; BNP, brain natriuretic peptide; CRP, C-reactive protein; MUST, malnutrition universal screening tool; ACE, angiotensin converting enzyme; ARB, angiotensin receptor blockers; MRA, mineralocorticoid receptor antagonist; DOACs; direct oral anticoagulants; SGLT2, sodium glucose transport protein 2.

**Table 2 jcm-10-05780-t002:** Clinical outcomes according to frailty status.

Outcome	Low Risk of Frailty HFRS < 5	Intermediate Risk of Frailty HFRS 5–15	High Risk of Frailty HFRS > 15	*p* Value
	*n* = 2830	*n* = 817	*n* = 59	
Overall				
LOS median (IQR)	4 (2.5, 6.7)	7.4 (4.7, 11.8)	14.5 (8.6, 20.6)	<0.001
In hospital mortality *n* (%)	98 (3.5)	118 (14.4)	22 (37.3)	<0.001
30-day mortality *n* (%)	164 (5.8)	188 (23.0)	34 (57.6)	<0.001
180-day mortality *n* (%)	390 (13.8)	299 (36.6)	42 (71.2)	<0.001
30-days readmissions *n* (%)	545 (19.3)	147 (17.9)	15 (25.4)	0.329
NH placement *n* (%)	137 (4.8)	83 (10.2)	9 (15.3)	<0.001
<65 years				
LOS median (IQR)	4.6 (2.8, 6.9)	9.8 (5.6, 14.2)	22.0 (20.6, 22.5)	<0.001
In hospital mortality *n* (%)	8 (1.3)	9 (8.9)	0	<0.001
30-day mortality *n* (%)	14 (2.3)	11 (10.9)	1 (20.0)	<0.001
180-day mortality *n* (%)	36 (6.0)	20 (19.8)	2 (40.0)	<0.001
30-days readmissions *n* (%)	99 (16.6)	20 (19.8)	1 (20.0)	0.713
NH placement *n* (%)	2 (0.3)	0	0	0.837
65–89 years				
LOS median (IQR)	4.0 (2.5, 6.7)	7.8 (5.1, 13.2)	15.2 (14, 17.6)	<0.001
In hospital mortality *n* (%)	20 (2.42)	31 (14.2)	4 (44.4)	<0.001
30-day mortality *n* (%)	34 (3.8)	40 (18.3)	6 (66.7)	<0.001
180-day mortality *n* (%)	93 (10.3)	61 (27.9)	7 (77.8)	<0.001
30-days readmissions *n* (%)	152 (16.8)	32 (14.6)	2 (22.2)	0.650
NH placement *n* (%)	12 (1.3)	8 (3.6)	1 (11.1)	0.009
≥80 years	3.9 (2.4, 6.4)	6.9 (4.2, 10.7)	13.4 (6.2, 19.1)	<0.001
LOS				
In hospital mortality *n* (%)	70 (5.3)	78 (15.7)	18 (40.0)	<0.001
30-day mortality *n* (%)	116 (8.7)	137 (27.6)	27 (60.0)	<0.001
180-day mortality *n* (%)	261 (19.6)	218 (43.9)	33 (73.3)	<0.001
30-days readmissions *n* (%)	294 (22.1)	95 (19.1)	12 (26.7)	0.262
NH placement *n* (%)	123 (9.3)	75 (15.1)	8 (17.8)	0.001

HFRS, Hospital Frailty Risk Score; LOS, length of hospital stay; IQR, interquartile range; NH, nursing home.

## Data Availability

The data presented in this study are available on request from the corresponding author only after permission is granted by the ethics committee.

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
