# Peer review of "Trends in Frailty and Use of Evidence-Based Pharmacotherapy for Heart Failure in Australian Hospitalised Patients: An Observational Study"

_jcm, 2021, doi:10.3390/jcm10245780_

Round 1
Reviewer 1 Report
Trends in frailty and use of evidence-based pharmacotherapy for heart failure in Australian hospitalised patients: an observational study
Reviewer comments
Authors aim to investigate the association between frailty and heart failure prescriptions.
The authors simplify frailty assessment only by applying administrative hospital data that previously demonstrated acceptable comparison to other clinical traditional scores.
The Hospital Frailty Assessment Tool uses administrative data in which previous admissions play an important role. Nonetheless, this kind of data not always reflect the accurately the frailty of the patient.
One wonders if the tool only provides marginal improvements in the prediction of clinical outcomes, is it worth the time?
Authors recognize they did not have the echocardiographic data to separate patients with preserved and reduced ejection fraction. Nonetheless they compare heart failure specific medications (part of which are only used in patients with reduced ejection fraction) between frail and non-frail patients. I think the ejection fraction can be used as a confounding factor in the analysis.
The authors stay “a significant proportion of patients younger than 65 years met HFRS criteria for frailty and similar to older counterparts, frailty was associated with worse outcomes in the younger age group “ I think a comparison between groups of frail and non-frail in older and younger would be a good idea in order to estimate the burden of frailty respect to outcomes in each group.
Author Response
Authors aim to investigate the association between frailty and heart failure prescriptions.
The authors simplify frailty assessment only by applying administrative hospital data that previously demonstrated acceptable comparison to other clinical traditional scores.
The Hospital Frailty Assessment Tool uses administrative data in which previous admissions play an important role. Nonetheless, this kind of data not always reflect the accurately the frailty of the patient.
One wonders if the tool only provides marginal improvements in the prediction of clinical outcomes, is it worth the time?
Response: We agree with reviewer’s comments, and it seems this tool provides marginal improvements in the predication of clinical outcomes. A future study which can add variables such as polypharmacy, BMI and physiological parameters to the HFRS may be useful to see if this can increase its prediction power.
Authors recognize they did not have the echocardiographic data to separate patients with preserved and reduced ejection fraction. Nonetheless they compare heart failure specific medications (part of which are only used in patients with reduced ejection fraction) between frail and non-frail patients. I think the ejection fraction can be used as a confounding factor in the analysis.
Response: We agree with reviewer’s comments and have included this as a major limitation of this study.
“We were unable to procure echocardiogram results of our patients and hence were unable to distinguish the impact of frailty in patients with heart failure with reduced and preserved ejection fractions. Nonetheless, the severity of heart failure in these patients was judged from the BNP levels [26].”
The authors stay “a significant proportion of patients younger than 65 years met HFRS criteria for frailty and similar to older counterparts, frailty was associated with worse outcomes in the younger age group “ I think a comparison between groups of frail and non-frail in older and younger would be a good idea in order to estimate the burden of frailty respect to outcomes in each group.
Response: We have now compared clinical outcomes in frail and non-frail patients in older (≥65 years) and younger patients (<65 years) and found worse clinical outcomes in terms of LOS, 30-day and 180-day mortality but not 30-day readmissions. We have now clarified this in the results section on page 7 and in Table 2.
“Frailty was associated with worse clinical outcomes in terms of LOS, 30-day and 180-day mortality in both younger patients (<65 years) and older patients (≥65 years) when compared to those who were classified as non-frail (P<0.05). When compared to non-frail patients, only those frail patients who were over the age of 65 years were more likely to be admitted to a nursing home (P<0.001). However, 30-day readmissions were not significantly different according to the categorisation of frailty by the HFRS in different age groups (Table 2).”
Reviewer 2 Report
What follows are my comments regarding the manuscript entitled “Trends in frailty and use of evidence-based pharmacotherapy for heart failure in Australian hospitalized patients: an observational study”, by Y. Sharma et al.
In this study, the authors reviewed the data from patients admitted to two Australian teaching hospitals from 2015 until 2019 with diagnosis of heart failure and examined the prevalence and influence of frailty and the prescription of medications for heart failure. The assessment of frailty was made with the Hospital-Frailty-Risk-Score (HFRS). The authors found that frail patients received less guideline-directed heart failure therapy. Moreover, they found a higher incidence of frail patients over time, despite no change in the average age of the patients.
The study addresses an issue that is not very “appealing” for publication, but very important. We commonly have to evaluate patients with CHF who are in a marginal state due to co-morbidities and very poor functional capacity. Methodology used in the manuscript is sound and the data has clinical relevance. Tables and figures are simple and informative. Of course, the study suffers of the usual limitations related to any retrospective study, nonetheless its data should be of value to clinicians evaluating patients with heart failure.
Author Response
Response: We thank reviewer for comments.